# Clinical Significance of Serum Lactate in Acute Myocardial Infarction: A Cardiac Magnetic Resonance Imaging Study

**DOI:** 10.3390/jcm10225278

**Published:** 2021-11-13

**Authors:** Ik Hyun Park, Hyun Kyu Cho, Ju Hyeon Oh, Woo Jung Chun, Yong Hwan Park, Mirae Lee, Min Sun Kim, Ki Hong Choi, Jihoon Kim, Young Bin Song, Joo-Yong Hahn, Seung-Hyuk Choi, Sang-Chol Lee, Hyeon-Cheol Gwon, Yeon Hyeon Choe, Woo Jin Jang

**Affiliations:** 1Department of Internal Medicine, Samsung Changwon Hospital, Sungkyunkwan University School of Medicine, Changwon 51353, Korea; ionoeval@gmail.com (I.H.P.); nara2002go@naver.com (H.K.C.); ojh@korea.com (J.H.O.); saintjmn@naver.com (W.J.C.); hippomac@hanmail.net (Y.H.P.); conatuse@gmail.com (M.L.); kjen9108@gmail.com (M.S.K.); 2Samsung Medical Center, Division of Cardiology, Department of Medicine, Heart Vascular Stroke Institute, Sungkyunkwan University School of Medicine, Seoul 06351, Korea; cardiokh@gmail.com (K.H.C.); novolley@naver.com (J.K.); youngbin.song@gmail.com (Y.B.S.); ichjy1@gmail.com (J.-Y.H.); cardiochoi@skku.edu (S.-H.C.); sc.lea@samsung.com (S.-C.L.); hcgwon@naver.com (H.-C.G.); 3Department of Radiology, Cardiovascular Imaging Center, Samsung Medical Center, Sungkyunkwan University School of Medicine, Seoul 06351, Korea; yh.choe@samsung.com; 4Division of Cardiology, Department of Internal Medicine, Seoul Hospital, Ewha Womans University College of Medicine, Seoul 07804, Korea

**Keywords:** acute myocardial infarction, serum lactate, percutaneous coronary intervention, magnetic resonance imaging

## Abstract

Little is known about causality and the pathological mechanism underlying the association of serum lactate with myocardial injury in patients with acute myocardial infarction (AMI). We evaluated data from 360 AMI patients undergoing percutaneous coronary intervention (PCI) using cardiovascular magnetic resonance imaging (CMR). Of these, 119 patients had serum lactate levels > 2.5 mmol/L on admission (high serum lactate group), whereas 241 patients had serum lactate levels ≤ 2.5 mmol/L (low serum lactate group). We compared the myocardial infarct size assessed by CMR between the two groups and performed inverse probability of treatment weighting (IPTW). In CMR analysis, myocardial infarct size was significantly greater in the high serum lactate group than in the low serum lactate group (22.0 ± 11.4% in the high serum lactate group vs. 18.9 ± 10.5% in the low serum lactate group; *p* = 0.011). The result was consistent after IPTW adjustment (21.5 ± 11.1% vs. 19.2 ± 10.4%; *p* = 0.044). In multivariate analysis, high serum lactate was associated with larger myocardial infarct (odds ratio 1.59; 95% confidence interval 1.00–2.51; *p* = 0.048). High serum lactate could predict advanced myocardial injury in AMI patients undergoing PCI.

## 1. Introduction

Acute myocardial infarction (AMI) is the most frequent cause of cardiogenic shock, and patients with AMI readily exhibit increased serum lactate levels [1,2]. Because serum lactate levels reflect hemodynamic status, and increased serum lactate indicates decreased systemic oxygen delivery or tissue hypoperfusion, serum lactate is a prognostic marker in critically ill patients in various forms of shock [2,3]. Previous studies have demonstrated the prognostic role of serum lactate and suggested point-of-care measurement of serum lactate at admission for evaluating hemodynamic status and risk stratification in AMI patients [3,4]. Vermeulen et al. [3] reported that higher lactate levels were independently associated with 30 day mortality, and Chebl et al. [5] also reported that hospital mortality was the highest in those with a lactate > 4.0 mmol/L, followed by those with a lactate >2.5–4.0 mmol/L and a lactate < 2 mmol/L (47.4% vs. 26.5% vs. 19.6%; *p* < 0.0001), suggesting the negative correlation of serum lactate with clinical outcomes among critically ill patients. However, although several studies have shown an association between serum lactate level and clinical outcomes, the causality and pathological mechanisms underlying the association remain poorly understood. Cardiovascular magnetic resonance imaging (CMR) can assess the extent of myocardial injury, providing valuable insight into the effects of serum lactate level in AMI patients [6,7]. Therefore, we evaluated an association between elevated levels of serum lactate and myocardial injury assessed by CMR in AMI patients undergoing percutaneous coronary intervention (PCI).

## 2. Materials and Methods

The study population consisted of patients on the Acute Myocardial Infarction–Cine Magnetic Resonance Imaging registry at Samsung Medical Center in Seoul, Korea, from December 2007 to July 2016. Inclusion criteria for this study were as follows: (1) patients treated with PCI for AMI and (2) patients who underwent CMR after the index procedure. Exclusion criteria were as follows: (1) a history of previous myocardial infarction, (2) medical conditions that could affect serum lactate levels, such as muscle injury or disease, diabetic ketoacidosis, hepatitis, suspected sepsis or other causes of shock, and cancer, and (3) missing information regarding serum lactate or poor-quality CMR for analysis. Finally, a total of 360 patients were included in this study (Figure 1). The Institutional Review Board of Samsung Medical Center approved this study, and all participants provided written informed consent to participate in this study.

Serum lactate level was measured from the study patients upon admission before any therapy. Since there is no known definitive lactate level to predict myocardial injury, and previous studies have suggested various baseline lactate levels, from 1.8 to 6.5 mmol/L as independent prognostic factors [3,8,9], we obtained the best threshold value of serum lactate (2.5 mmol/L) to predict large myocardial infarct. Based on this value, study patients were divided into two groups: those with serum lactate levels > 2.5 mmol/L (high serum lactate group) and those with serum lactate levels ≤ 2.5 mmol/L (low serum lactate group). We additionally stratified the study population into three subgroups according to lactate levels of 2.5 mmol/L and 4.0 mmol/L, to clarify the correlation between serum lactate and myocardial infarct size: a lactate ≤ 2.5 mmol/L group, a lactate > 2.5–4.0 mmol/L group, and a lactate > 4.0 mmol/L group [10,11].

The primary outcome was myocardial infarct size (% of left ventricle or ventricular (LV)) assessed by CMR according to baseline serum lactate levels. Secondary outcomes were extent of the area at risk (AAR; % of LV), myocardial salvage index (MSI), and microvascular obstruction (MVO) area (% of LV).

Research coordinators of the dedicated registry prospectively recorded baseline characteristics, angiographic findings, and CMR data. Blood samples to determine serum lactate and creatine kinase myocardial band (CK-MB) were drawn from patients at admission. Serum CK-MB level was measured every eight hours from the index procedure until the peak value was confirmed. Baseline left ventricular ejection fraction (LVEF; %) was measured by transthoracic echocardiography using Simpson’s method just after PCI. AMI was defined as evidence of myocardial injury (elevation of cardiac troponin values, with at least one value above the 99th percentile upper reference limit) with necrosis in a clinical setting, consistent with myocardial ischemia [11]. Multi-vessel disease was defined as stenosis > 50% in more than two coronary arteries. Thrombolysis in myocardial infarction flow grade and myocardial blush grade were evaluated using the final angiogram, as defined previously [12]. All baseline and procedural cine coronary angiograms were reviewed and analyzed quantitatively at the angiographic core laboratory of our institution.

CMR was performed using a 1.5-T scanner (Achieva, Philips Medical Systems, Best, Netherlands). All measurements were performed at the Samsung Medical Center–CMR core laboratory using validated software (ARGUS; Siemens Medical System, Erlangen, Germany). Infarct size and extent of MVO were assessed using late gadolinium-enhanced images, whereas AAR was measured on T2-weighted images [13]. Two experienced radiologists who were blinded to patient information performed measurements using the software described above. Endocardial borders were traced after acquiring short-axis images at end-diastole and end-systole. The Simpson algorithm was used to calculate LV end-diastolic volume, LV end-systolic volume, and LVEF. Infarct size was calculated as summation of the area with delayed hyperenhancement within each segment of the short-axis images. This value was multiplied by slice thickness to cover the entire LV. Endocardial and epicardial borders were planimetered to calculate the myocardial area, which was summed to calculate LV myocardial volume using the same method. Infarct size was expressed as percentage of affected LV myocardial volume. T2-weighted images were used to determine the presence of hemorrhagic infarction [14]. AAR was quantified on T2-weighted images using a similar algorithm to that described above and was expressed as percentage of LV myocardial volume affected. MSI was computed as follows: MSI = (AAR-infarct size)/AAR ×100 [15].

Continuous variables were expressed as mean ± standard deviation or median when they lacked a normal distribution. Analysis of continuous variables was performed using Student’s *t*-test or Wilcoxon rank-sum test. Categorical variables were described as number (n) with percentage (%), and differences were analyzed by Pearson χ^2^ or Fisher’s exact tests. We performed inverse probability of treatment weighting (IPTW) adjustment to reduce treatment selection bias according to treatment strategy and other potential confounding factors. The IPTW method was performed using generalized boosted models for baseline characteristics to evaluate if interaction affected clinical outcomes. Covariates that were either statistically significant in univariable analysis (*p* value < 0.1) or considered clinically important were included in multivariate models to determine the independent predictors of large myocardial infarcts (percentage infarct volume ≥ mean infarct size in the present study). The analyzed covariates were age, sex, body mass index, hypertension, diabetes mellitus, ST-segment elevation myocardial infarction (STEMI), multi-vessel disease, and serum lactate. All tests were two-tailed, and *p* < 0.05 was considered statistically significant. Statistical analyses were performed using SAS software (Version 9.2; SAS Institute Inc., Cary, NC, USA).

## 3. Results

### 3.1. Baseline Clinical, Angiographic, and Procedural Characteristics

A total of 360 patients were divided into the high serum lactate group (*n* = 119; 33%) and the low serum lactate group (*n* = 241; 67%) according to the best cut-off value of baseline serum lactate of 2.5 mmol/L (Figure 1). The mean lactate levels of the high and low serum lactate groups were 4.3 ± 2.5 mmol/L and 1.6 ± 0.5 mmol/L, respectively. There was higher incidence of males (*p* = 0.018) and diabetes mellitus (*p* = 0.001) in the high serum lactate group than the low serum lactate group. Compared with the low serum lactate group, the high serum lactate group had significantly higher white blood cell count (*p* <0.001) and peak CK-MB (*p* = 0.002). Other demographic and clinical characteristics were not significantly different between the two groups (Table 1). Table 2 shows the angiographic and procedural findings in the high and low serum lactate groups. Excluding the presence of collateral flow (44.3% in the high serum lactate group vs. 58.0% in the low serum lactate group; *p* = 0.018), angiographic and procedural findings were not significantly different between the two groups (Table 2).

### 3.2. Analysis of CMR Findings

CMR was performed a median of 3.5 days (interquartile range, 2.3–4.2) after the index procedure, and intervals from primary PCI to CMR were not significantly different between the two groups. Myocardial infarct size was significantly greater in the high serum lactate group than the low serum lactate group (22.0 ± 11.4% vs. 18.9 ± 10.5%; *p* = 0.011). The extent of AAR was significantly larger in the high serum lactate group than in the low serum lactate group (37.4 ± 17.7 vs. 32.1 ± 15.1; *p* = 0.003). MSI (41.1 ± 16.6 vs. 41.6 ± 16.8; *p* = 0.780) and MVO area (3.9 ± 5.7 vs. 3.6 ± 5.4; *p* = 0.597) were similar between the two groups (Table 3). After IPTW adjustment, the differences in myocardial infarct size (21.5 ± 11.1 vs. 19.2 ± 10.4; *p* = 0.044) and extent of AAR (37.0 ± 17.2 vs. 32.7 ± 15.0; *p* = 0.013) were consistently significant between the two groups. MSI (41.5 ± 16.8 vs. 41.8 ± 16.8; *p* = 0.865) and MVO area (3.7 ± 5.5 vs. 3.6 ± 5.4; *p* = 0.680) were similar between the two groups (Table 3 and Figure 2).

### 3.3. Correlation between Serum Lactate and Myocardial Infarct Size

To clarify the association between serum lactate and myocardial infarct size, the high serum lactate group was divided into two subgroups according to serum lactate level of 4.0 mmol/L [9,10]. Finally, the study population was stratified into three groups: a lactate ≤2.5 mmol/L group (*n* = 241; 67%), a lactate >2.5–4.0 mmol/L group (*n* = 80; 22%), and a lactate >4.0 mmol/L group (*n* = 39; 11%), and CMR findings were compared between the three groups. The higher serum lactate group showed significantly larger myocardial infarct size (18.9 ± 10.5 in the lactate ≤2.5 mmol/L group vs. 21.5 ± 11.9 in the lactate >2.5–4.0 mmol/L group vs. 22.9 ± 10.4 in the lactate >4.0 mmol/L group; *p* = 0.031) and AAR size (32.1 ± 15.1 vs. 35.7 ± 17.9 vs. 40.8 ± 17.0; *p* = 0.003) than the lower serum lactate group. There were no significant differences in MSI (41.6 ± 16.8 vs. 40.1 ± 16.6 vs. 42.9 ± 16.7; *p* = 0.664) and MVO area (3.6 ± 5.4 vs. 3.9 ± 5.6 vs. 4.0 ± 5.9; *p* = 0.860) between the three groups. After IPTW adjustment, consistently significant differences were observed in myocardial infarct size (19.2 ± 10.5 vs. 20.8 ± 11.7 vs. 23.5 ± 10.6; *p* = 0.009) and AAR size (32.6 ± 15.0 vs. 35.1 ± 17.7 vs. 42.1 ± 17.1; *p* < 0.001) between the three groups. MSI (41.7 ± 16.8 vs. 41.0 ± 16.5 vs. 43.1 ± 16.3; *p* = 0.607) and MVO area (3.6 ± 5.4 vs. 3.8 ± 5.4 vs. 4.5 ± 6.9; *p* = 0.491) were not significantly different between the three groups after IPTW adjustment (Figure 3 and Figure A1). IPT-adjusted baseline, angiographic, and procedural characteristics of the three groups are presented in Table A1 and Table A2.

### 3.4. Predictors of Large Myocardial Infarct

In multivariate logistic regression analysis, independent predictors of large myocardial infarct (percent infarct volume ≥ 20%) were baseline serum lactate >2.5 mmol/L (odds ratio (OR) 1.59; 95% confidence interval (CI) 1.00–2.51; *p* = 0.048), body mass index (OR 0.60; 95% CI 0.39–0.94; *p* = 0.024), and STEMI (OR 2.22; 95% CI 1.4–3.55; *p* = 0.001) (Figure 4).

## 4. Discussion

The association between serum lactate level and myocardial injury was investigated in patients with AMI using CMR markers of myocardial and microvascular damage. The main finding of this study is that in AMI patients undergoing PCI, high serum lactate is associated with greater myocardial infarct size and extent of AAR, as assessed by CMR. In multivariate analysis, high serum lactate was associated with larger myocardial infarct, and the association between high serum lactate and advanced myocardial injury was consistent across various subgroups. To the best of our knowledge, this is the first study to evaluate the clinical significance of high serum lactate level on myocardial injury as assessed by CMR data in AMI patients. Our findings correspond well with those of earlier studies that established an association between elevated serum lactate and adverse clinical outcomes [5,16]. Therefore, the result of the present study may provide the causality and pathological mechanisms for the association of adverse clinical outcomes with high serum lactate in patients with AMI.

Serum lactate has been studied over time, and its prognostic role in patients with sepsis or shock has been reported in many studies [1,2,3,17]. Serum lactate is an important prognostic predictor in AMI patients, but limited data are available regarding its clinical significance, causality, and pathological mechanism. Slottosch et al. reported that baseline serum lactate was predictive of 30 day mortality, but they only analyzed a specific population of shock patients supported with ECMO [1]. Lazzeri et al. [8] reported that serum lactate is a prognostic marker for early mortality in patients with STEMI, and Fuernau et al. [18] showed that serum lactate measured at eight hours after shock onset had predictive value but could not elucidate an underlying mechanism. CMR is positioned uniquely to evaluate comprehensively the morphological, functional, and microvascular sequelae of AMI patients [7] and can be used to assess almost all relevant prognostic pathophysiological consequences of myocardial ischemia and reperfusion after AMI [13]. We evaluated the association of serum lactate on admission and prognostic pathophysiology of myocardial ischemia in patients undergoing PCI for AMI using CMR data, and we identified high serum lactate to be associated with extensive myocardial and microvascular damage. Because our findings suggest that elevated serum lactate should be understood not only as a phenomenon secondary to tissue hypoxia or hypoperfusion, but also as a predictor of larger myocardial injury, they provide substantial support that high serum lactate is a reasonably accurate surrogate for advanced myocardial injury in AMI patients. Furthermore, the results can explain the higher incidence of adverse clinical outcomes reported by Lazzeri et al. and Fuernau et al. [8,18]. We used the best threshold value of serum lactate to predict large myocardial infarct size. Our threshold value of 2.5 mmol/L is a significant independent predictor of in-hospital mortality as well as a tool for assessing underlying hypoperfusion [10]. Since the prevalence of hemodynamic deterioration after 24 h increases with higher Society for Cardiovascular Angiography and Intervention (SCAI) shock stages [19], a cut-off value of 2.5 mmol/L on admission can be used to discriminate patients in need of early intervention (inotrope, vasopressor support, or mechanical circulatory support) beyond volume resuscitation to restore perfusion and could be a surrogate marker of successful therapy in AMI patients [20]. Therefore, our results strengthen the understanding that blood lactate is one of the most precise serum markers, and that serum lactate testing should be performed more frequently on critically ill patients, including those with AMI [2]. Because high serum lactate might be deleterious to the jeopardized myocardium, close monitoring of serum lactate and appropriate treatment for AMI can reduce the expansion of myocardial damage and adverse clinical outcomes. However, these are hypothesis-generating findings, and the potential therapeutic implications of these findings deserve further investigation.

A sub-analysis of CMR data was conducted according to stratified serum lactate level to clarify the correlation between serum lactate and myocardial injury. Previously, Chebl et al. [5] stratified study patients into three groups according to lactate levels of 2.0 mmol/L and 4.0 mmol/L to determine if serum lactate was associated independently with mortality among critically ill patients. In their study, serum lactate levels of 2.0 to 4.0 mmol/L and >4.0 mmol/L were independent predictors of in-hospital mortality, but the studied population was heterogeneous. To elucidate the negative correlation of serum lactate with myocardial injury, we further divided the high serum lactate group (those with serum lactate >2.5 mmol/L) into two groups based on a serum lactate concentration of 4.0 mmol/L and compared myocardial injury between the three subgroups. Patients with higher serum lactate levels had significantly greater myocardial infarct size, larger extent of AAR, and lower LV ejection fraction compared with patients with lower serum lactate, which confirmed that serum lactate level parallels an increase in myocardial injury. This finding also explains the results of previous studies that reported a negative correlation between serum lactate and clinical outcomes [10,11]. Because high serum lactate represents more active inflammation and hypoxic injury, which could play an important role in the immune response that mediates atherosclerosis [21], more active inflammation in patients with high serum lactate might result in larger myocardial infarct. Interestingly, we also found that the extent of myocardial injury had a negative relationship with high body mass index (≥25 kg/m²). This corresponds well with findings of earlier studies that established an association between obesity and clinical outcomes (obesity paradox) in patients with AMI [22,23,24,25], and the results of our study might provide the causality and pathological mechanisms.

The current study evaluated the prognostic role of baseline serum lactate in patients with AMI undergoing PCI. The strength of our study is that we investigated the pathophysiological aspects of serum lactate that explain the underlying mechanism of adverse clinical outcomes, as assessed by CMR, unlike previous studies that analyzed the association between high serum lactate and adverse clinical outcomes. We analyzed the prognostic role of lactate according to stratified serum lactate level to clarify the negative relationship of serum lactate with myocardial damage and found that higher serum lactate was significantly more likely to cause greater myocardial damage. Our findings imply that early intensive therapy based on risk stratification using close serum lactate monitoring may improve clinical outcomes.

This study has several limitations. First, its design was nonrandomized, prospective, and observational, which might have significantly affected results attributed to confounding factors. Second, a patient undergoing CMR might be clinically stable with modest myocardial injury. Because only patients available for CMR were included, the sample size of our study was small, which may limit our study results. Third, due to the retrospective nature of our registry, we could not thoroughly identify the detailed data regarding clinical outcomes including mortality, and the impact of serum lactate on assessment of outcomes was not assessed in this study. Fourth, we chose a high level of serum lactate based on previous studies regarding the clinical meanings of serum lactate, although there are several definitions of serum lactate in real-world practice [16]. Thus, the statistical association of high serum lactate with findings on CMR might change according to high lactate definitions. Finally, although various examinations were performed to exclude sepsis or septic shock, there is a chance that patients with unidentified infection may be included in the study. The non-infectious or non-cardiogenic causes of high serum lactate, such as excessive muscle activity including seizure, diabetic ketoacidosis, liver dysfunction, medication side effects, trauma, and cancer, were not fully evaluated.

## 5. Conclusions

High serum lactate is associated with larger myocardial infarct size and greater extent of myocardial edema, with a negative correlation observed, as assessed by CMR. Based on our results, serum lactate level could be a prognostic factor in patients with AMI, and close attention needs to be paid to AMI patients to reduce expansion of myocardial injury, especially those with baseline serum lactate >2.5 mmol/L. Further investigation regarding the potential therapeutic implications of these findings should be reported.

## Figures and Tables

**Figure 1 jcm-10-05278-f001:**
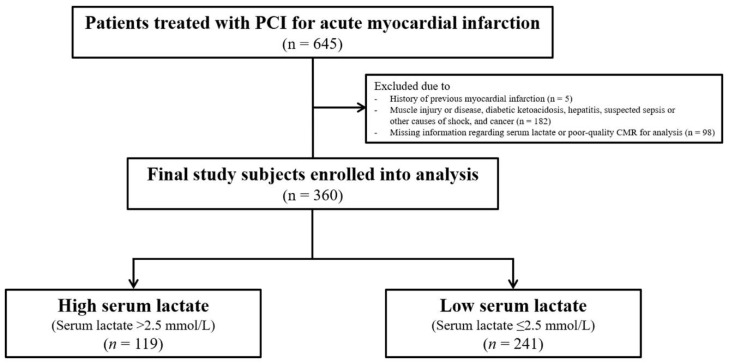
Schematic illustration of study cohort selection. PCI = percutaneous coronary infarction.

**Figure 2 jcm-10-05278-f002:**
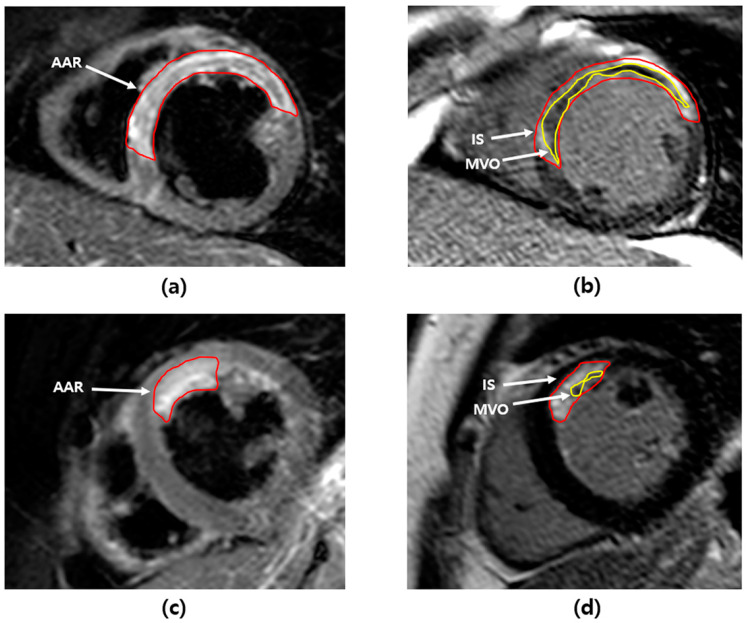
Cardiac magnetic resonance images of acute myocardial infarction. (**a**) AAR (55.2%) in T2-weighted image and (**b**) IS (44.8%) and MVO (10.4%) in late gadolinium hyperenhancement image of a patient in the high serum lactate group. (**c**) AAR (12.3%) in T2-weighted image and (**d**) IS (8.8%) and MVO (4.3%) in late gadolinium hyperenhancement image of a patient in the low serum lactate group. AAR = area at risk, IS = infarct size, MVO = microvascular obstruction.

**Figure 3 jcm-10-05278-f003:**
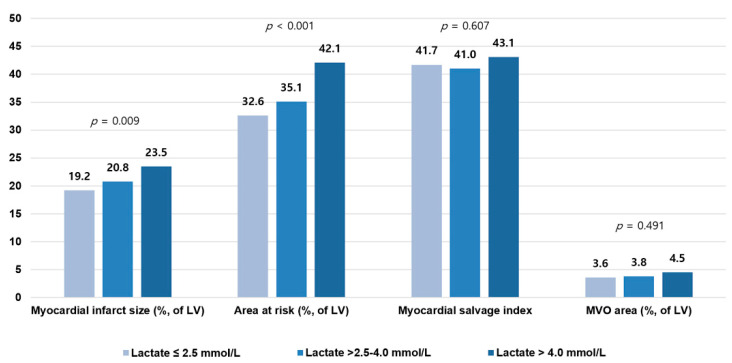
Relation between serum lactate levels and cardiac magnetic resonance findings. Bar plots show the relation between baseline serum lactate levels and myocardial infarct size, area at risk, myocardial salvage index, and MVO area. MVO = microvascular obstruction. See Table A1 and Table A2.

**Figure 4 jcm-10-05278-f004:**
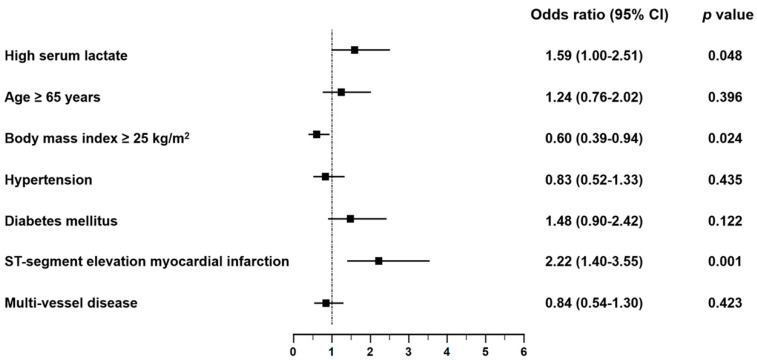
Predictors of large myocardial infarct. Forest plots show the results of multivariate analysis of predictors of large myocardial infarct. CI = confidence interval.

**Table 1 jcm-10-05278-t001:** Baseline characteristics.

	Overall Population (*n* = 360)			IPTW Population (*n* = 718)		
	High Serum Lactate	Low Serum Lactate	*p* Value	High Serum Lactate	Low Serum Lactate	*p* Value
	(*n* = 119)	(*n* = 241)		(*n* = 358)	(*n* =360)	
Age	61.5 ± 13.0	59.6 ± 11.4	0.180	59.8 ± 13.4	60.1 ± 11.4	0.763
Male	89 (74.8)	205 (85.1)	0.018	292 (81.4)	394 (81.6)	0.973
Body mass index, kg/m²	24.2 ± 4.0	24.8 ± 3.3	0.124	24.8 ± 4.3	24.6 ± 3.3	0.574
Current smoker	45 (37.8)	106 (44.0)	0.265	153 (42.8)	150 (41.7)	0.832
Hypertension	61 (51.3)	104 (43.2)	0.146	165 (46.1)	166 (46.1)	>0.999
Diabetes mellitus	44 (37.0)	51 (21.2)	0.001	98 (27.3)	97 (27.0)	0.942
Dyslipidemia	19 (16.0)	51 (21.2)	0.241	55 (15.2)	80 (22.1)	0.096
History of percutaneous coronary intervention	13 (10.9)	17 (7.1)	0.211	30 (8.3)	33 (9.1)	0.793
History of cerebrovascular accident	8 (6.7)	6 (2.5)	0.078	27 (7.4)	10 (2.7)	0.040
Clinical presentation						
NSTEMI	35 (29.4)	85 (35.3)	0.267	120 (33.5)	121 (33.5)	>0.999
STEMI	84 (70.6)	156 (64.7)	0.267	238 (66.5)	239 (66.5)	>0.999
Laboratory findings						
NT-proBNP, pg/mL	240.0 (66.4–1098.8)	186.4 (51.4–862.9)	0.126	198.7 (57.3–849.4)	196.9 (53.6–926.3)	0.618
White blood cell, ×10^3^/μL	15.7 ± 6.5	11.5 ± 3.5	<0.001	15.4 ± 6.0	11.5 ± 3.5	<0.001
Hemoglobin, g/dL	14.0 ± 1.9	14.4 ± 1.9	0.062	14.3 ± 1.9	14.3 ± 2.0	0.876
Platelet ×10^3^/μL	237.6 ± 65.6	229.4 ± 60.1	0.240	236.2 ± 60.2	231.6 ± 61.3	0.472
Peak CK-MB, ng/mL	143.1 (40.6–293.2)	88.6 (29.2–214.2)	0.002	148.6 (56.2–287.4)	90.7 (28.5–214.1)	<0.001
**Concomitant medications**						
Aspirin	117 (98.3)	239 (99.2)	0.602	353 (98.6)	356 (99.0)	0.729
P2Y12 inhibitor	117 (98.3)	237 (98.3)	1.000	352 (98.3)	354 (98.4)	0.943
Beta blocker	104 (87.4)	216 (89.6)	0.526	313 (87.4)	325 (90.3)	0.385
ACE inhibitor or ARB	93 (78.2)	193 (80.1)	0.670	283 (79.0)	287 (79.7)	0.867
Statin	112 (94.1)	228 (94.6)	0.849	339 (94.6)	342 (95.0)	0.853

Data are presented as *n* (%), mean ± standard deviation, or median (interquartile range). ACE = angiotensin-converting enzyme, ARB = angiotensin receptor blocker, CK-MB = creatine kinase-myocardial band, IPTW = inverse probability of treatment weighting, NT-proBNP = N-terminal prohormone of brain natriuretic peptide, STEMI = ST-segment elevation myocardial infarction, NSTEMI = non-ST-segment elevation myocardial infarction.

**Table 2 jcm-10-05278-t002:** Angiographic and procedural characteristics.

	Overall Population (*n* = 360)			IPTW Population (*n* = 718)		
	High Serum Lactate	Low Serum Lactate	*p* Value	High Serum Lactate	Low Serum Lactate	*p* Value
	(*n* = 119)	(*n* = 241)		(*n* = 358)	(*n* =360)	
Infarct-related artery			0.782			0.635
Left anterior descending artery	58 (48.7)	113 (46.9)		161 (44.8)	165 (45.8)	
Left circumflex artery	18 (15.1)	36 (14.9)		55 (15.4)	54 (15.1)	
Right coronary artery	43 (36.1)	90 (37.3)		142 (39.8)	137 (38.2)	
Left main artery	0 (0.00)	2 (0.8)		0 (0.00)	3 (0.9)	
Multi-vessel disease	56 (47.1)	121 (50.4)	0.549	175 (48.9)	177 (49.1)	0.976
TIMI flow grade before PCI			0.988			0.921
0	81 (68.1)	161 (66.8)		249 (69.5)	244 (67.9)	
1	6 (5.0)	14 (5.8)		17 (4.8)	23 (6.4)	
2	16 (13.4)	34 (14.1)		48 (13.5)	49 (13.5)	
3	16 (13.4)	32 (13.3)		44 (12.3)	44 (12.2)	
Final TIMI flow grade 3 after PCI	107 (90.7)	214 (92.6)	0.523	320 (89.4)	322 (89.5)	0.346
Angiographic no reflow phenomenon	7 (5.9)	14 (6.1)	0.962	20 (5.5)	22 (6.2)	0.746
Presence of collateral flow	51 (44.3)	127 (58.0)	0.018	164 (45.9)	191 (52.9)	0.063
Myocardial blush grade			0.367			0.450
0	2 (1.7)	1 (0.5)		7 (1.9)	1 (0.4)	
1	1 (0.9)	3 (1.4)		4 (1.2)	5 (1.3)	
2	15 (13.0)	18 (8.6)		43 (12.0)	30 (8.3)	
3	97 (84.3)	188 (89.5)		290 (81.0)	281 (78.1)	
Aspiration thrombectomy	65 (54.6)	115 (47.7)	0.218	191 (53.4)	172 (47.8)	0.291
Use of GPIIb/IIIa inhibitor	14 (12.2)	36 (17.1)	0.235	41 (11.4)	55 (15.2)	0.157
Number of implanted stents	1.21 ± 0.58	1.15 ± 0.36	0.591	1.2 ± 0.5	1.3 ± 0.7	0.126
Stent diameter (mm)	1.2 ± 0.5	1.3 ± 0.7	0.288	3.1 ± 0.5	3.1 ± 0.4	0.929
Stent length (mm)	30.2 ± 15.7	32.9 ± 17.6	0.193	30.3 ± 15.2	33.3 ± 18.1	0.113

Data are presented as *n* (%), mean ± standard deviation. GP = glycoprotein, IPTW = inverse probability of treatment weighting, PCI = percutaneous coronary intervention, TIMI = thrombolysis in myocardial infarction.

**Table 3 jcm-10-05278-t003:** Analysis of cardiac magnetic resonance findings.

	Overall Population (*n* = 360)			IPTW Population (*n* = 718)		
	High Serum Lactate	Low Serum Lactate	*p* Value	High Serum Lactate	Low Serum Lactate	*p* Value
	(*n* = 119)	(*n* = 241)		(*n* = 358)	(*n* = 360)	
Myocardial infarct size (%, of LV)	22.0 ± 11.4	18.9 ± 10.5	0.011	21.5 ± 11.1	19.2 ± 10.4	0.044
Area at risk (%, of LV)	37.4 ± 17.7	32.1 ± 15.1	0.003	37.0 ± 17.2	32.7 ± 15.0	0.013
Myocardial salvage index	41.1 ± 16.6	41.6 ± 16.8	0.780	41.5 ± 16.8	41.8 ± 16.8	0.865
MVO area (%, of LV)	3.9 ± 5.7	3.6 ± 5.4	0.597	3.7 ± 5.5	3.6 ± 5.4	0.680
Hemorrhagic infarction	53 (44.9)	96 (39.8)	0.359	166 (46.3)	143 (39.8)	0.194
LV end diastolic volume (mL)	149.4 ± 42.3	151.7 ± 40.9	0.623	151.2 ± 40.2	150.8 ± 41.2	0.915
LV end systolic volume (mL)	77.6 ± 39.6	75.1 ± 36.8	0.557	77.0 ± 36.1	75.3 ± 36.9	0.663
LV ejection fraction (%)	50.1 ± 11.8	52.3 ± 11.2	0.090	50.7 ± 11.0	51.8 ± 11.3	0.354
LV stroke volume (mL)	71.8 ± 17.3	76.6 ± 16.7	0.011	74.2 ± 17.7	75.5 ± 16.8	0.489
LV cardiac output (L/min)	5.1 ± 1.1	5.2 ± 1.1	0.669	5.2 ± 1.1	5.1 ± 1.1	0.573

Data are presented as *n* (%) or mean ± standard deviation. IPTW = Inverse probability of treatment weighting, LV = left ventricle (ventricular), MVO = microvascular obstruction.

## Data Availability

The data presented in this study are not available due to legal restrictions.

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
