# Peer review of "Clinical Significance of Serum Lactate in Acute Myocardial Infarction: A Cardiac Magnetic Resonance Imaging Study"

_jcm, 2021, doi:10.3390/jcm10225278_

Round 1
Reviewer 1 Report
Dear Authors,
Please add more conclusions to your research
Please extent your introduction
Kind regards
Reviewer 2 Report
The manuscript by Park et al has defined the relation of serum lactate with myocardial infraction. The manuscript is well written and results are supported by the data. I have the following comments to improve the manuscript.
- Please provide good images of CMR echo data to show the infarcted region in compared groups and include them in the main manuscript.
- Please provide a good paragraph on the translational outlook of the study and its clinical importance.
